# Online Pilot Grocery Intervention among Rural and Urban Residents Aimed to Improve Purchasing Habits

**DOI:** 10.3390/ijerph19020871

**Published:** 2022-01-13

**Authors:** Alison Gustafson, Rachel Gillespie, Emily DeWitt, Brittany Cox, Brynnan Dunaway, Lindsey Haynes-Maslow, Elizabeth Anderson Steeves, Angela C. B. Trude

**Affiliations:** 1Department of Dietetics and Human Nutrition, University of Kentucky, Lexington, KY 40506, USA; brittany.cox@uky.edu (B.C.); brynnan.jacobs@uky.edu (B.D.); 2Department of Family and Consumer Sciences Extension, University of Kentucky, Lexington, KY 40506, USA; rachel.gillespie@uky.edu (R.G.); emily.dewitt@uky.edu (E.D.); 3Agricultural & Human Sciences, North Carolina State University, Raleigh, NC 27695, USA; lhaynes-maslow@ncsu.edu; 4Department of Nutrition, University of Tennessee, Knoxville, TN 37996-1920, USA; eander24@utk.edu; 5Department of Nutrition and Dietetics, Steinhardt School of Culture, Education, and Human Development, New York University, New York City, NY 10003, USA; angela.trude@nyu.edu

**Keywords:** online, grocery shopping, behavioral nudge, intervention, rural, urban, fruit and vegetable, food access

## Abstract

Online grocery shopping has the potential to improve access to food, particularly among low-income households located in urban food deserts and rural communities. The primary aim of this pilot intervention was to test whether a three-armed online grocery trial improved fruit and vegetable (F&V) purchases. Rural and urban adults across seven counties in Kentucky, Maryland, and North Carolina were recruited to participate in an 8-week intervention in fall 2021. A total of 184 adults were enrolled into the following groups: (1) brick-and-mortar “BM” (control participants only received reminders to submit weekly grocery shopping receipts); (2) online-only with no support “O” (participants received weekly reminders to grocery shop online and to submit itemized receipts); and (3) online shopping with intervention nudges “O+I” (participants received nudges three times per week to grocery shop online, meal ideas, recipes, Facebook group support, and weekly reminders to shop online and to submit itemized receipts). On average, reported food spending on F/V by the O+I participants was USD 6.84 more compared to the BM arm. Online shopping with behavioral nudges and nutrition information shows great promise for helping customers in diverse locations to navigate the increasing presence of online grocery shopping platforms and to improve F&V purchases.

## 1. Introduction

Prior to the COVID-19 pandemic, in 2019 online grocery sales grew 22% relative to 2018 in the United States (US). After COVID-19 cases were confirmed in the US, severe closures and a surge in online grocery shopping (including the delivery of items ordered online and pick-up at store location of food ordered online) for various food and beverages, with an increase of 48% in online sales was observed [1]. There was a record high of USD 5.3 billion in online sales in April of 2020, with continued growth in May [2]. Yet, rural customers, and those participating in the supplemental nutrition assistance program (SNAP), still report barriers to online grocery ordering, including delivery fees, inconvenient pick-up times, and an overall lack of availability of online grocery services in their geographic area [3,4,5]. Recent evidence suggests a limited uptake of online grocery shopping, especially among rural populations, even when financial incentives are provided [6,7]. However, there are strong indicators that those who shop online spend less overall, purchase less sugary snacks and candies, and purchase more fruits and vegetables [8,9,10,11,12,13]. Online grocery shopping has a strong potential to improve food access and dietary intake. Thus, in order to help with the unmodifiable structural (delivery access, internet capacity [8]) barriers, interventions can be implemented to help customers to overcome the more modifiable barriers (such as unfamiliarity with the online ordering websites), such as reminders to shop online to maintain consistency, and recipes to help with setting up the grocery cart online for healthier purchases.

The 2014 farm bill mandated a pilot study to test the feasibility and implications of allowing retail food stores to accept SNAP electronic benefits transfer (EBT) through online transactions, with customers being allowed to make online purchases using their SNAP EBT benefits at authorized retailers [6]. The initial mandate was aimed to first test the feasibility of a secure and safe online benefit redemption. After testing, the SNAP online capacity began to expand, during the COVID-19 surge, to other stores beyond Amazon and Walmart, which would provide extensive reach for many low-income households [6]. SNAP online is now available in 47 states across a wide variety of retailers. This is a critical opportunity for the digital marketplace to expand their online ordering functions, such as behavioral prompts to improve healthy choices, to encourage the utilization of digital coupons, and to enhance the comfort of using the online ordering functions, with the intent to improve food purchases for low-income customers [14].

To date, lower-income residents are less likely to shop online for food relative to higher-income households [8]. There are also limited online delivery options in rural communities and fewer stores in rural communities offering online delivery [4,7]. In addition, there is a risk with unguided access to online grocery shopping as increased exposure to less healthy options could exacerbate diet-related disease [4]. Yet, there is strong evidence that online shopping can help to decrease impulse purchases [15], improve purchases of fruits and vegetables [10], and improve food security among lower-income residents [16].

Research suggests that the expansion of online shopping in lower-income communities with nutrition education may address food insecurity and improve dietary quality [10]. Online grocery shopping has a strong potential to decrease impulse purchases typically conducted in brick-and-mortar stores [15]. Specifically, online shopping through pre-filled grocery cart “nudges”, nutrition education prompts, and nutrition labeling may reduce impulse purchases, such as chips, snacks, and high-calorie foods, while improving purchases of fruits, vegetables, and whole grains, relative to shopping at a brick-and-mortar store [8,9,10,12]. A recent study indicated the strong potential for rural households utilizing online shopping to increase the overall quality of foods purchased [10]. Customers need assistance to help them become better acquainted with online ordering capabilities and to overcome several reported barriers to improve what healthy items are added to the online grocery cart. Several SNAP interventions conducted in grocery stores indicated that shoppers make more nutritious choices when multiple nudges are utilized. Specifically, a systematic review indicated that behavioral economic strategies, such as nudges of easy to understand quick nutrition information, improve purchases of fruits and vegetables [17]. Several studies using choice architecture constructs, such as changing the store layout and the prominent positioning of healthy foods, improved purchases of healthier foods among those customers [5,17,18,19,20]. Lastly, a recent study using nudges for online shoppers has indicated that virtual shopping trials using nudges and price incentives improved the purchases of healthy foods for low-income consumers [21]. These findings provide evidence that utilizing nudges as a person orders their food online may help to improve what is placed in their grocery cart.

However, there are limited intervention trials testing how online shopping may improve total purchases of fruits and vegetables among diverse geographic and socioeconomic samples. Thus, the study authors have utilized several of the SNAP grocery shopping interventions and tailored them for use in online shopping.

To the authors knowledge, this study is the first to conduct a pilot quasi-experimental intervention among rural and urban shoppers designed to test the effectiveness of an intervention across three study arms. The aim of the study was to test whether the intervention achieved the following: (1) improved modifiable self-reported barriers to online shopping and (2) improved average weekly amount of fruits and vegetables purchased.

## 2. Materials and Methods

### 2.1. Study Setting

The intervention took place in Kentucky (KY), Maryland (MD), and North Carolina (NC), across seven counties. Three counties were in rural KY and NC, and four counties were in urban NC, KY, and MD. Counties were selected based on rural-urban continuum codes (3–8) with the aim of representing both urban and rural settings; grocery stores offering online ordering; and Cooperative Extension buy-in for community-based recruitment efforts.

Inclusion and exclusion Criteria—Participant eligibility requirements included adults aged 21 and older that were the primary shopper in the household, spoke English as their primary language, had a cell phone that could receive text messages, agreed to conduct online shopping, had a phone or device that allowed the ordering of food online, and agreed to participate in the intervention for 8 weeks. Exclusion criteria included individuals that indicated that they did not live in the county were recruitment was conducted, reported a severe chronic disease that would alter their purchases, were pregnant, or were planning on becoming pregnant.

### 2.2. Enrollment and Randomization

There were two phases of enrollment between February and July 2021. The first phase included Facebook advertisement posts to each corresponding study county’s Cooperative Extension page with an EZ Text number that interested participants could text to learn more about the study and enroll (EZ Text is a mobile app that offers free texting services, which were overseen by the study team.) This resulted in *n* = 204 eligible participants. The next phase consisted of setting up information tables with local health departments and Cooperative Extension at grocery stores frequented by residents of the selected counties, which resulted in an additional *n* = 52 eligible individuals. There was not a specific income criteria or SNAP enrollment. However, recruitment was conducted in rural counties with high poverty rates, high SNAP percentage, and among stores that accept SNAP online. Additionally, it should be noted that during this study period, SNAP eligibility and benefits were expanded to cover more individuals at a greater amount of funding levels. Therefore, the study authors were not as concerned with income verification. The enrollment consisted of individuals completing the electronic consent form, followed by a baseline survey conducted via phone. Those who completed the baseline survey received a USD 50 Mastercard gift card for participation, by mail. A total of *n* = 183 individuals were enrolled in the online grocery intervention (1:1:1 randomization ratio). A computer-generated randomization was used among rural and then urban residents given the fixed effect of online shopping options among rural shoppers. Thus, simple randomization was used with stratification using computer randomization [22].

Incentive structure—All participants received a USD 50 gift card at the beginning of the intervention after completing the baseline survey, another USD 10 per week for 8 weeks after sending in their receipts, and another USD 50 at the conclusion of the 8-week intervention upon completion of the post-intervention survey. The gift card was a Mastercard gift card from the Western Union-University of Kentucky pilot program. A total of USD 10 was uploaded each week to their Mastercard automatically as an incentive to turn in their receipts and to help defray any costs associated with online grocery shopping (i.e., delivery fees).

Retention—After four weeks, *n* = 49 participants stopped participating or opted out and were therefore removed from the study. After seven weeks, an additional *n* = 5 participants stopped participating or opted out and were removed from the study. These removal time periods were outlined to participants in the consent form, indicating that individuals may be removed from the study if they did not respond or participate in study activities for three consecutive weeks. The final sample after eight weeks with pre- and post-surveys and two weeks of receipts was *n* = 129. See Figure 1 for study design and enrollment across study arms.

### 2.3. Study Design

Individuals were randomized into online-only or online + intervention arms within each county. Those that did not have access to online shopping were placed in the control group “BM” (*n* = 13). Since availability of online grocery service depended on the retailer business model, the effect of online shopping was “fixed”. When online grocery services were available in the county, households were randomized into one of the three study arms. However, to maintain a balance between urban and rural residents across the study arms the three arms are not exactly balanced. In addition, to be able to make comparisons for price and availability across store types, only three large supermarket retailers were used for online shopping within each state. Kentucky utilized counties that had Food City, Kroger, and Walmart. Maryland utilized an urban county with Kroger chains. While North Carolina utilized counties that had Harris Teator (a Kroger subsidiary), Food Lion (similar to Food City), and Walmart. Thus, this was a quasi-experimental study, as counties could not be completely randomized to receive online shopping or not, and residents were randomized within their counties. Residents in the seven counties were enrolled into one of three study arms, as follows: (1) brick-and-mortar (BM)—continue grocery shopping as they normally do; (2) online-only (O)—no assistance with messages or healthy shopping, however, weekly text messages were sent to encourage online shopping; and (3) online + intervention (O+I)—weekly nudges were sent to assist with healthy meal planning, recipes, and to continue online grocery shopping three times per week.

### 2.4. Intervention Components

After enrollment, all participants were mailed a welcome packet that explained how to redact receipts and submit them weekly to the study team, along with information on the importance of keeping the Mastercard gift card for use throughout the study. Those in the O arm also received instructions given in the mailed welcome packet for how to set up their shopping cart online, while those in the O+I arm received information about when they would receive reminder text messages and how to join the private Facebook group for meal ideas and recipes.

Brick-and-mortar group (*n* = 55)—participants in this arm only received text message reminders to submit their food shopping receipts by sending pictures of their receipts via text or by submitting weekly through pre-paid envelopes to receive USD 10 loaded to their Mastercard gift card. No behavioral nudge messaging was sent directly to the participants, but individuals were prompted to continue their engagement in the project through weekly shopping reminders.

Online-only group (*n* = 45)—In week one, participants were provided with a welcome packet to help them set up their online grocery cart. After the online carts were created, participants received a behavioral nudge, which comprised specific language, each week on Saturday to renew their carts for the following week. Participants were encouraged to send images of their cart or receipts after they had placed their grocery shopping order to receive USD 10 loaded onto their Mastercard gift card.

Online+Intervention group (*n* = 29)—In week one, participants were provided with a welcome packet to help them set up their online grocery cart. In the subsequent weeks, behavioral nudges and prompts were sent to participants three times per week. In addition, a private Facebook group page was set up to help with social connection between members of this study arm. The overall content was structured around the following modifiable barriers to online shopping: (1) perceptions that food is more expensive online; (2) reminders to set up their carts to avoid inconvenient pick-up times; and (3) prompts to help navigate ordering groceries online to decrease technology barriers related to online grocery shopping platform functions. Based on previous research about barriers to online shopping [15,23], the behavioral nudges focused on meal planning, meal preparation, reminders to set up their online grocery cart each week, strategies to stretch their food dollars, and choosing fruit and vegetable items that were seasonal and more affordable. The Facebook posting mimicked the content from the text messages but also included similar content from the Plate It Up! Kentucky Proud University of Kentucky Cooperative Extension program [24].

Text messaging schedule—BM group participants received the same text message every Saturday reminding them to submit their shopping receipts. The O group participants received a text message every Saturday that varied in nature, although provided a behavioral nudge to continue to shop online. These messages were motivational, specific if needed, and tailored to the location of the participant (KY, NC, or MD). The O+I group participants received a text message three times a week, which included several behavioral nudges to assist with healthy meal planning, online shopping, recipes, and motivation to improve self-efficacy with online grocery shopping and making healthy shopping choices. The O+I group participants were also invited to join a private Facebook group that offered more recipes and meal planning tips with resources. Three Facebook posts were delivered to members each week in addition to the weekly text messages.

Among those who were not responding to messages, further individual-level tailored prompts were sent to maintain engagement in the intervention. Figure 2 depicts an example image that was created and posted for the O+I Facebook group as a behavioral nudge for meal planning. In addition to the behavioral nudges, a text was sent each Saturday to remind participants to set up their grocery cart. Listed below is an example of a text message nudge that was sent to those in the O+I arm of the study (full content of messages available upon request), as follows:

“Start your day off right with a tasty breakfast! Try a simple egg scramble with veggies you have leftover, or a yogurt parfait with your favorite fruit. Eating breakfast can give you the energy to tackle the day ALL day!”

The general content of the messages was previously tested and validated in a grocery store intervention among rural residents living counties with high rates of poverty, obesity, food insecurity, and poor diet quality [24]. The content of the messages has been utilized in previous studies, showing acceptability, efficacy, and modifying behavior [25]. However, the exact wording was derived by research assistants based on four primary principles, (1) supporting self-efficacy, (2) providing simple and affordable recipes, (3) utilizing affective language, and (4) offering encouragement and motivational language to support positive behaviors. The study team based the text messages on “Nudge Theory”, which indicates that low-cost text messages can have broad applications to guide a healthier lifestyle. As coined by Thaler and Sunstein, the authors suggest that there exists a “choice architecture”, which involves outside forces that guide decisions [26]. The outside force in this intervention was the physical grocery store but also the online platform where food choices were being made when items were placed into the shoppers’ online grocery cart. Results from a meta-analyses on nudge interventions indicated that, on average, a nudge resulted in a 15% increase in healthier consumption [27]. Thus, the study is grounded in nudge theory and is utilizing this approach through healthy text messages being sent at crucial “choice” moments when shopping online.

### 2.5. Measures

Baseline and post-intervention surveys, including preferences and barriers to online shopping, were provided after informed consent was obtained, and participants were contacted to reserve a time for baseline survey data collection. Trained graduate students in the Department of Dietetics and Human Nutrition at the University of Kentucky conducted the survey via phone at the baseline among *n* = 183 participants. The post inte-vention survey was conducted among the *n* = 129 participants who completed the intervention. The survey collected information on demographics (age, race/ethnicity), general shopping habits, and online shopping preferences and barriers. The online shopping preferences and barriers questions were used from previous research [15,23,28] as well as key collaboration across study authors.

Text messaging process evaluation—All text messages sent to participants encouraged them to respond. The text message exchange between the research team and participants was tracked weekly to assess study engagement for process evaluation purposes according to the study arm messaging schedule. Engagement and fidelity were measured separately and collected depending on arm of study. Engagement was defined as a response to the weekly text message within 24 h across BM and O study arms. Each text message sent by the research team that the participant did not reply to was coded as “0”, a single text back from the participants was coded as “1”, and a multiple text exchange was coded as “2”. Messages that included receipts were not counted in the process evaluation.

Facebook process evaluation—Social media metrics for the Facebook group (O+I participants only), were measured. Metrics collected included reach, dose delivered, and fidelity indicators, as these have been collected and measured in previous studies using social media and are proven to adequately track intervention implementation quality [29]. Reach was defined as the number of times the post was viewed by individual page followers. Dose delivered was defined as the number of total posts and messages that were sent per week by the research team. Fidelity was defined as the measure of engagement on a post, which included number of ‘likes’, comments, or replies generated from the post, plus responses from weekly text messages. Type of message delivered was also collected to assess engagement levels dependent on content shared (e.g., recipes, meal planning or cooking tips, motivational or affective messaging). Individuals that were in O+I and were not part of the Facebook group (i.e., did not have a personal Facebook page or chose not to join the group) were tracked as ‘missing’, whereas participants who were in the group and did not interact received a ‘zero’ for each post.

Primary outcomes (F&V purchases) were assessed by collecting itemized grocery receipts from participants weekly. Participants were encouraged to submit receipts for all foods purchased for consumption at home and as described above, were given USD 10 each week that receipts were collected [30,31]. Participants either mailed in their receipts using pre-stamped envelopes from all their food venues where food would be consumed at home, or they took screen shots of their receipts and texted or e-mailed them to the research team.

Receipt coding—All receipts collected from participants were analyzed to determine fruit and vegetable purchase, subtotal of receipt (total amount spent before taxes), total amount spent on fruits and vegetables per receipt, and percentage of receipt spent on fruits and vegetables relative to the total amount spent was then calculated. Fruits and vegetables included any fresh, frozen, or canned fruits and vegetables, as well as vegetable soups. Condiment type foods, such as olives and pickles, were not included and salsa and tomato sauce were also not included. The list of foods that we included was based on USDA-ARS fruit and vegetable categories “What We Eat in America Food Categories 2017–2018” (https://www.ars.usda.gov/ARSUserFiles/80400530/pdf/1718/Food_Category_List.pdf, accessed on 6 January 2022). Receipts that indicated ‘medley’ in the frozen section were assumed as vegetable medley and were counted towards total fruit and vegetable purchases.

Type of shopping coding—Receipts were coded as online or in-store, based on the receipt indicating cashier for in-store purchases or online, to examine percentage spent of food from different grocery platforms. Next, receipts were coded as first trip (in-store vs. online) to indicate their primary food shopping trip of the week. Participants then submitted additional receipts when subsequent food shopping trips were conducted. All of these receipts ended up being in-store and thus were coded as second shopping trip.

The University of Kentucky Institutional Review Board approved this study (IRB Protocol #61763).

### 2.6. Data Analysis

Descriptive statistics were derived using means, percentages, and chi-square to compare differences across study arms. Power calculation (Table 1) indicates that *n* =128 is needed for an effect size of 0.25%, at 80% power to declare that the mean of the paired differences is significantly different from zero. To model the change in purchasing habits over 8 weeks, panel data was established. Xtreg was used to set panel data in Stata 16.0 (StataCorp. 2019; StataCorp LLC, College Station, TX, USA). GLM with fixed effects and instrumental variable for rural/urban was used in all models. Instrumental variable was used based on relevance, exclusion, and exchangeability. Given that our sample had fixed exposure to online shopping vs. in-store and those in rural communities vs. urban communities are systematically different, we tested the rural/urban variable using two-stage least squares estimator [32]. This variable was then used as the IV in primary outcome analyses. Models with total fruit and vegetable purchase and online controlled for the total bill. Sensitivity analyses were conducted between the sample that dropped out or were removed for now having complete receipt data relative to the full enrolled sample. There were no significant differences in gender, age, education, or income between those that dropped out or were removed from study relative and those that stayed in the intervention. However, there are unobservable characteristics that are unaccounted for among our small sample. There is a strong possibility that those who dropped out of the study differ systematically to the sample that remained in the intervention. Thus, results need to be interpreted with caution.

## 3. Results

### 3.1. Baseline Characteristics and Purchasing Findings

Table 2 details descriptions of the study participants collected at the baseline and indicate that participants were predominantly female. The mean age range across the study arms was 38–46 years, with a majority of residents having lived in their county for 10 years or more (range of 62%–75%), and over half of participants were college graduates (range of 54%–65%). There were no significant differences between race, income, or education across the study arms at the baseline. However, there was a significant difference across study arms between the rural and urban status. Although every attempt was made to randomize across the study arms, there was a significant difference between the arms on the rural and urban status. All primary outcome analyses used an instrumental variable to account for these differences across the arms.

Although the study participants in the BM arm were encouraged to shop in-store for their food at home purchases, 13% of purchases were still made online. This may be due to ordering food from Amazon and picking up food ordered online close to their workplace. Those in the O and O+I arms were encouraged to shop online. However, 60% of food purchased among those in the O arm were conducted online, and 65% of foods purchased among the O+I arm was conducted online. There was a significant difference between shopping habits across the study arms (*p* = 0.001) with those in the O+I shopping more online relative to the BM arm. The mean total bill among the BM arm was USD 128.39 (SE 5.69), while those in the O arm spent on average USD 115.25 (SE 7.08), and those in the O+I arm spent USD 116.54 (SE 7.11). These averages are slightly higher than the lowest income quintile of spending USD 80 per week on food, but similar to the second income quintile of spending USD 115 per week on food [33].

There were no significant differences in mean purchases across the study arms. We did not capture if food ordered online was from pick-up or delivery, thus results present online orders from pick-up or delivery.

### 3.2. Fruit and Vegetable Purchases and Shopping Type

Table 3 presents the results for the primary outcome of total spent on fruit and vegetable purchases, in addition to the total grocery bill. There were no significant differences across the study arms over the 8 weeks for the average grocery total bill (spent both online and in-store) or the average total amount spent online. However, those in the O+I study arm spent, on average, USD 6.84 (95% CI 3.58–10.11) more on fruits and vegetables compared to the BM arm.

The results from the secondary analyses related to type of shopping (online vs. in-store) on total bill and total fruit and vegetables purchases is reported in Table 4. As shown in Table 2, a significant percentage of shoppers conducted both online and in-store shopping. Thus, our analyses addressed participants who conducted their first shopping trip of the week online compared to those who conducted their first shopping trip of the week in-store. The results indicate that those who shopped online for their first trip of the week spent, on average, USD 3.80 more on fruits and vegetables compared to those who shopped in-store for the first trip. There were no significant differences for any other outcomes.

### 3.3. Online Shopping Attitudes and Barriers

At the baseline, there were no significant differences between the study arms for strengths related to price, quality of food available online, online availability of foods people like to consume, access to internet, delivery options, and online shopping saving time (Table 5). There were no significant differences at the baseline between the study arms for the barriers to online shopping related to online websites being difficult to use, searching for product labels taking too long, online pickup times being inconvenient, delivery fees making participants less likely to order, and minimum purchase fees acting as a barrier to online shopping.

The post-intervention results displayed in Table 5 show (1) change in the self-reported strengths to online shopping between the baseline and post-intervention, and (2) differences post-intervention across the study arms. First, there was a significant change between the baseline and post-intervention for food prices being affordable online. Those in the O arm at the baseline reported agreeing or strongly agreeing with affordability and at post-intervention there was a significant change in participants disagreeing about affordability. Second, of those in the BM arm, 77% (*n* = 24) indicated that they agreed or strongly agreed that the quality of food items is good online. Conversely, only 39% (*n* = 9) of those in the O arm and 36% (*n* = 4) of those in the O+I arm agreed or strongly agreed. Of those in the BM arm, 25% (*n* = 8), indicated that they agreed or strongly agreed that food items are available online that they like to purchase. While 73% (*n* = 19) of those in the O arm and 63% (*n* = 5) of those in the O+I arm agreed or strongly agreed.

The post-intervention results indicate (1) changes in the self-reported barriers to online shopping between the baseline and post-intervention, and (2) differences post-intervention across the study arms. First, there were significant differences post-intervention across the study arms for the following barriers to online shopping: the online site being difficult to use, searching for labels taking too long, online pickup times being inconvenient, and minimum purchasing requirements acting as barriers to online shopping. Second, there were significant differences between the baseline and post-intervention for the following barriers to online shopping: the online site being difficult to use, searching for labels taking too long, online pickup times being inconvenient, and delivery fees making the person less likely to order. In general, those in the O and the O+I arms reported disagreeing or strongly disagreeing with the barriers to online shopping.

Given the small sample size and the dropout among the participants, we also report on the overall online shopping experience among those who stayed in the study. The feedback was solicited from the participants via EZ Text and Facebook by asking for comments or suggestions for their stores to improve the online ordering process. Those who responded provided insight into perceptions of affordability. Participants shared the following statements:

“It is pretty convenient plus I noticed it save me money because I don’t see things and throw in my buggy like I do in the store.” -O+I participant (rural)

“I think the online ordering is good. Prices are pretty accurate to the instore price on items. The one negative is sometimes the cold items could be colder.” -O participant (urban)

“I love how [store] has no minimum order for pick up. And using sale ad and planning meals saves money. A few times I haven’t been pleased with quality of the fruit. Small price to pay.” -O+I participant (urban)

### 3.4. Engagement and Fidelity Findings across Study Arms

Across the three study arms, engagement and fidelity were collected. Engagement was defined as a response to the weekly text message within 24 h across the study arms. Fidelity was defined as the measure of engagement on a Facebook post, which included number of ‘likes’, comments, or replies generated from the post, plus responses from weekly text messages among O+I participants only. Figure 3 details the engagement trends for all of the study arms. The engagement across the eight weeks of the intervention for the BM participants totaled 81, 74, 39, 42, 24, 42, 44, and 41 responses, respectively. The engagement across the eight weeks for the intervention period for the O study arm participants totaled 55, 53, 33, 51, 24, 32, 41, and 39 responses, respectively. The engagement across the eight weeks for the O+I study arm participants totaled 56, 46, 40, 25, 22, 23, 31, and 19 responses, respectively. The participants’ opt-out and removal rate (due to no receipts being sent) influenced the engagement across the 8-week study with the O+I group most impacted from week one to eight.

The Facebook metrics were collected, and an average was calculated for the three weekly posts throughout the eight weeks. On average, each post received 11.2 views, 1.3 likes, and 1.1 comments. The total reach for the Facebook group equaled 23 of a possible *n* = 55 participants each week, therefore *n* = 32 individuals in the O+I group opted not to join the Facebook group, did not have Facebook, or were removed from the study. The weekly views declined as the study progressed, beginning at 48 for week one of the study and concluding at 30 at week eight of the study. However, the total weekly posts were viewed on average 33.6 times by the participants.

The total fidelity as measured weekly for the O+I participants were as follows: 86, 70, 48, 42, 35, 26, 36, and 28, respectively.

## 4. Discussion

This study is the first pilot intervention to actively enroll participants into online shopping arms relative to brick-and-mortar. Although this study could not mandate that participants shop online for their food, our results point to how assisting customers with online grocery shopping can help to improve modifiable barriers to online shopping and, therefore, improve purchases of fruits and vegetables without increasing the overall total bill of the customers. Previous studies have cited barriers to online shopping, such as delivery fees, inconvenient pickup times, reluctance to purchase fresh produce online, cost, and distrust of the online ordering platform [3,23]. Although this study could not eliminate the delivery fees or the inconvenient pickup times, this intervention did target the barriers of distrust and reluctance of purchasing produce online through guided assistance and tailored nudges to help build social support around online shopping. The intervention also informs as to how assisting customers to navigate the online shopping space through meal planning tips, reminders to set up their online cart, weekly behavioral prompts related to recipes, and online sales, can help customers to effectively shop online and improve their purchases of fruits and vegetables. Coupled with the social media and text messaging components, this intervention led to stable engagement effectively encouraging both online grocery shopping, and healthy food purchasing behaviors. However, there was a larger drop-out among the O+I arm relative to other arms. This finding highlights how online shopping with nudges may provide a burden to certain types of shoppers. Future interventions should continue to explore longitudinal purchasing patterns in order to better understand consumer behavior and preferences when using online ordering platforms.

Our results are positioned within a limited but growing research field of online grocery shopping interventions [5,6,9,10,12,28,34]. To date, a few interventions have revealed how online shopping has helped to increase the purchase of high fiber foods [9] and decrease the purchase of less healthy food items that are high in saturated fats [12]. However, as mentioned previously, most of these studies tend to be simulation models, conducted pre-COVID-19 and the SNAP online pilot, and have less generalizability to actual customers’ shopping behaviors. Thus, our study built upon the previous research [5,10,15,28,34] to establish the content of the intervention to help improve future policy and public health practice applications aimed at assisting customers with online shopping. Although the data for all of the food purchases made during this study were not collected, these findings are encouraging when examining innovative strategies to improve food access, nutritional intake, and ultimately the health status of rural and SNAP populations, who generally have disproportional high rates of diet-related chronic diseases, in part due to nutritional inadequacy. Furthermore, previous studies have attempted to target these populations and improve purchasing habits utilizing behavior nudging principles to modify behaviors [17,29,35]. However, as online grocery shopping continues to grow as an engagement method, these principles can be applied to an online landscape rather than an in-store approach alone in an effort to improve purchases [21]. One benefit from using an online shopping study design is that once the digital infrastructure is in place, it can be more cost effective than in-person direct education and in-store behavioral nudges. Tailored strategies to support opportunities for online grocery shopping among these subpopulations will be impactful as these purchasing options become more widely available at additional stores across the US.

The growth predictions of online grocery shopping [2,36], in addition to the already existing online shopping options, has shed light on how the food environment as a whole is growing and evolving. Customers have even more ways to acquire food and, thus, research needs to keep pace with how customers are interacting with their food environment, and to make access to this type of shopping more equitable across geographic and socioeconomic differences [4,7,34]. There needs to be a concerted effort to understand and intervene within this food venue space in order to help consumers make healthful choices. Food venue options continue to increase but, if not equally distributed, can leave out marginalized subpopulations (rural, racial/ethnic populations) and widen existing disparities [4,7]. Thus, future research needs to examine these barriers and develop innovative ways to utilize online grocery shopping to promote healthier purchases across diverse populations. Future research needs to examine the reasons that participants to do not maintain online shopping behaviors in order to help industry and government tailor online platforms to meet the needs of customers in a healthful manner. Online shopping has the potential to decrease impulse purchases and provide a tailored shopping cart to help improve healthier purchases. A prime example of this growth is the predictive shopping cart being developed by Google and food corporation Albertsons. Research partners have a key role to help the industry to tailor these predictive models to promote healthier and affordable purchases over less healthy items. To date, several retailers have already begun to offer memberships for grocery delivery [37], while some third-party providers of grocery delivery have expanded their partnerships to include dollar-type stores, convenience stores, and other non-traditional food venues [38]. This is a prime opportunity for industry professionals, collaborators within public health, transportation, city planning, engineering, economics, marketing, and several other disciplines to partner in order to decrease disparities while exploring and expanding this scope of increased food accessibility and utilization of this grocery shopping method.

Our study is not without limitations. Although large efforts were made to maintain engagement in the study through weekly text messages, mailing of post-cards, and direct phone calls, our study had a 30% attrition rate over the 8 weeks (55/184 = 29.9%). However, relative to other interventions, this was a rather low attrition rate, which points to how nudges and various forms of engagement through texting can assist participants to stay active during the study duration. The study did not collect food purchases from every type of purchase (such as fast-food restaurants, gas stations, or farmers’ markets) and, thus, associations between the intervention and purchasing habits only reflect what participants chose to send via receipts [30,39]. The study authors were interested in understanding if the food purchased to be consumed at home changed over the 8 weeks. Thus, there is a limited understanding of whether or not online shopping also influenced food purchases away from home, such as in gas stations, fast-food, and traditional restaurants. There were sample size differences across the three groups, which can greatly impact the interpretation of the findings. Many of our participants lived in rural communities with limited broadband access and, thus, had limited ability to consistently order food online. There needs to be a concerted effort in policy changes moving toward to improve broadband access. There is limited information about which exact behavioral nudges worked specifically in this context. Thus, future work will be utilizing the multiphase optimization strategy (MOST) for larger-scale evaluation [40]. Lastly, there was no maintenance phase to determine whether or not these shopping habits persisted after the study ended.

## 5. Conclusions

This pilot study provides suggestive findings related to how online shopping may improve food shopping habits, however, results need to be confirmed with a larger, more rigorous study. This study helps to inform future research and policy to improve accessibility to food outlets that accept SNAP, and to better understand online grocery shopping practices among rural and urban residents [28,34]. As the growth and utilization of online grocery shopping persists, a unique opportunity is presented for several industries to partner in an effort to improve dietary outcomes among customers. A tailored experience that includes automatic place-based behavioral nudges and interactive nutrition information while customers are shopping online may help consumers to better navigate and utilize online grocery shopping services. This balance of open consumer choice with some regulation and crafting of online grocery landscapes, and communication could be a viable medium for policy makers to consider between the private and public sector.

## Figures and Tables

**Figure 1 ijerph-19-00871-f001:**
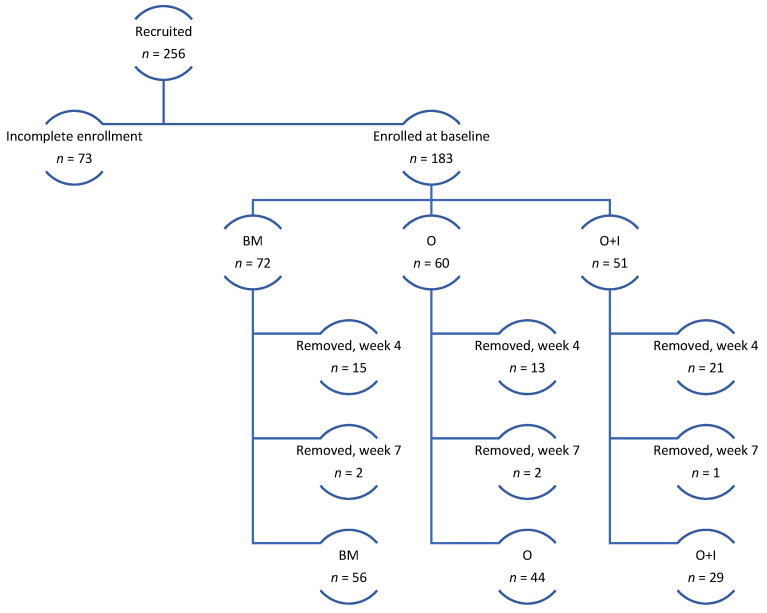
Study design and enrollment. BM = brick-and-mortar control; O = online-only; O+I = online+intervention content.

**Figure 2 ijerph-19-00871-f002:**
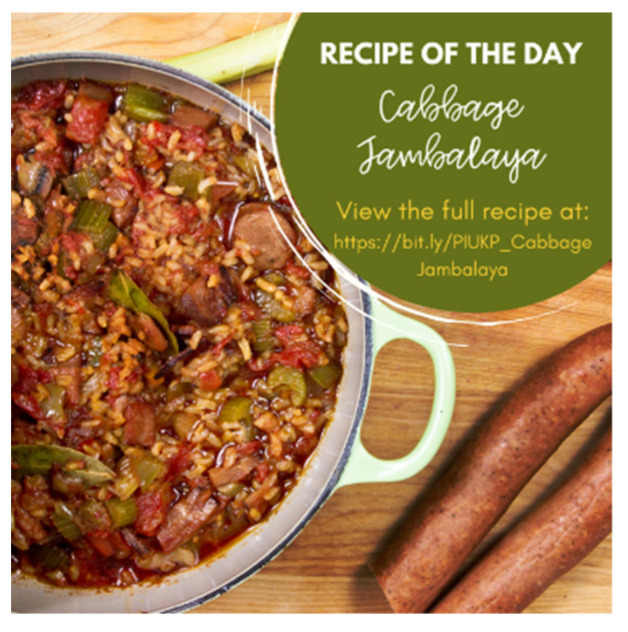
Example of a post for O+I Facebook group participants.

**Figure 3 ijerph-19-00871-f003:**
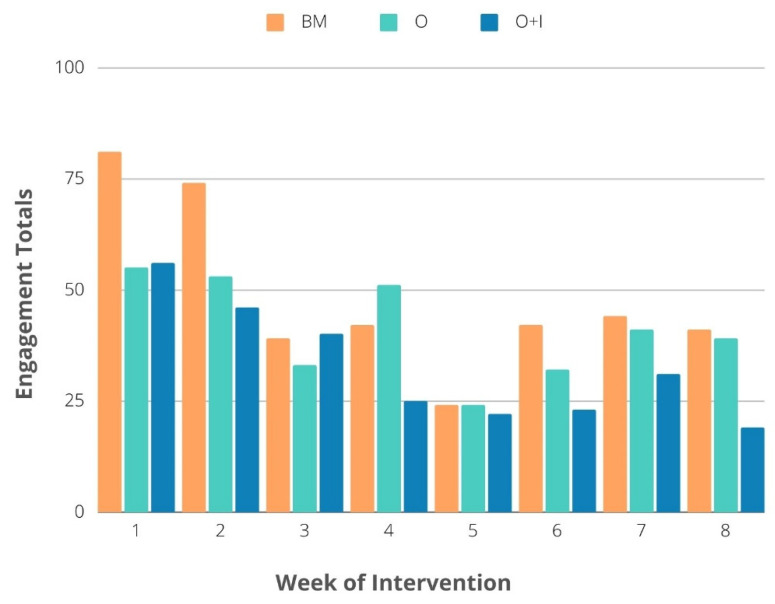
Engagement metrics across the three study arms for the eight-week intervention.

**Table 1 ijerph-19-00871-t001:** Sample Size Power Calculation.

Outcome	Alpha	Power	Proportion Difference between Control and Intervention	*n* Total
Purchase fruit and vegetables	0.05	0.8	0.25	128

128 is needed for an effect size of 0.25%, at 80% power to declare that the mean of the paired differences is significantly different from zero.

**Table 2 ijerph-19-00871-t002:** Demographics of study participants across study arms of intervention (*n* = 129).

Study Participant Descriptive ^1^	Brick-and-Mortar (*n* = 56)	Online-Only (*n* = 44)	Online + Message (*n* = 29)	*p*-Value
Gender				
Female	56 (100%)	42 (96%)	27 (96%)	0.237
Male	0	2 (4%)	2 (4%)	
Age (mean years-SD)	46 (1.59)	41 (1.48)	38 (1.85)	0.78
Length of Residence				0.42
10 years or less	25% (14)	31% (14)	38% (11)	
Greater than 10 years	75% (42)	68% (30)	62% (18)	
Education				0.336
High School or less	15 (27%)	5 (11%)	4 (13%)	
Some College	10 (18%)	11 (25%)	6 (21%)	
College Graduate	30 (54%)	28 (63%)	19 (65%)	
Race				0.62
White	45 (81%)	32 (72%)	20 (69%)	
Black or African American	9 (16%)	9 (20%)	7 (24%)	
Asian	1 (1%)	1 (2%)	1 (2%)	
Other	0 (0%)	2 (4%)	1 (3%)	
Household Income				0.30
Less than 20,000	12 (22%)	5 (11%)	2 (75%)	
21–49,000	16 (30%)	16 (37%)	11 (37%)	
50–69,999	13 (24%)	13 (30%)	5 (18%)	
70–99,999	10 (18%)	6 (13%)	5 (18%)	
Children in Household				0.2
No	27 (48%)	12 (27%)	10 (34%)	
1–2	21 (38%)	22 (50%)	16 (55%)	
3 or more	14 (25%)	17 (39%)	13 (44%)	
Supplemental Nutrition Assistance Program (SNAP)				0.169
Yes	16 (28%)	18 (41%)	6 (21%)	
No	40 (71%)	26 (59%)	23 (79%)	
Urban/Rural				0.002 *
Rural	23 (41%)	12 (27%)	20 (69%)	
Urban	33 (58%)	32 (72%)	9 (31%)	
BMI (mean SE)	33.49 (1.39)	32.69 (1.43)	35.99 (2.08)	0.69
Facebook				0.22
Daily	53 (94%)	36 (83%)	27 (94%)	
General Online Shopping Habits				0.27
Less than once a week	28 (50%)	20 (45%)	10 (34%)	
More than once a week	28 (50%)	24 (55%)	19 (65%)	
Purchasing Type (percentage that shopped in-store or online)				0.001 *
In-store	87%	40%	35%	
Online	13%	60%	65%	
Purchasing Habits (mean)				
Total Bill (in-store and online)	128.39 (5.69)	115.25 (7.08)	116.54 (7.11)	0.552
Total Bill Online	106.88 (12.07)	90.31 (6.48)	90.11 (5.78)	0.506
Total Bill In-store	83.91 (19.91)	79.99 (10.65)	91.65 (15.33)	0.51
Fruit and Vegetable Total (in-store and online)	9.67 (0.66)	12.27 (1.15)	16.23 (1.33)	0.26
Fruit and Vegetable Total Online	9.90 (1.45)	10.92 (1.16)	13.31 (1.34)	0.40

^1^ Means and percentages were derived using descriptive statistics. Chi-square was used to test for differences across categories. * Indicates significant differences across study arms (*p* < *0*.05).

**Table 3 ijerph-19-00871-t003:** Intervention effect on total purchases and purchases of fruits and vegetables across study arms.

Primary and Secondary Outcomes ^1^	Average across 8 Weeks
Total Bill (USD)	
Brick-and-mortar	Comparison
Online-only	−11.83 (−38.85, 15.19)
Online + Intervention	−14.78 (−39.66, 9.90)
Online Bill (USD)	
Brick-and-mortar	Comparison
Online-only	−3.45 (−45.61, 38.71)
Online + Intervention	11.55 (−38.69, 61.71)
In-store Bill (USD)	
Brick-and-mortar	Comparison
Online-only	−15.75 (−55.36, 23.86)
Online + Intervention	4.36 (−36.44, 45.16)
Total F/V purchases (USD)	
Brick-and-mortar	Comparison
Online-only	3.12 (-.46, 6.72)
Online + Intervention	6.84 (3.58, 10.11) *
Online purchases of F/V (USD)	
Brick-and-mortar	Comparison
Online-only	1.58 (−3.71, 6.88)
Online + Intervention	3.34 (−2.05, 8.73)

^1^ xtreg was used to set panel data in Stata. GLM with fixed effects and instrumental variable for rural/urban was used in all models. Models with total fruit and vegetable purchase and online controlled for total bill. * Indicates *p* < 0.05 with robust standard errors. F/V = fruits and vegetables.

**Table 4 ijerph-19-00871-t004:** Purchase Type—Association between how food was purchased online compared to in-store [reference] on total bill and fruit/vegetable bill.

Primary and Secondary Outcomes ^1^	Average across 8 Weeks
Total Bill (both online and in-store purchases)	1.22 (−20.81, 23.36)
Online-only Bill	12.60 (−17.35, 42.55)
In-store Only Bill	−50.03 (−201.47, 101.35)
Total fruit and vegetable purchases (both online and in-store purchases)	3.80 (1.21, 6.40) *
Online purchases of fruits and vegetables	0.24 (−5.79, 6.27)

^1^ xtreg was used to set panel data. GLM with fixed effects and instrumental variable for rural/urban was used in all models. Models with total fruit and vegetable purchase and online purchases of fruits and vegetables controlled for total bill. The first row is the beta coefficient followed by 95% CI. * Indicates *p* < 0.05 with robust standard errors.

**Table 5 ijerph-19-00871-t005:** Online shopping attitudes and barriers baseline and post-intervention across study arms.

Attributes of Online Shopping	Shopping Attitudes ^1^	Baseline			Difference at Baseline between Study Arms	Post-Intervention			Difference Post-Intervention between Study Arms	Difference Between Baseline and Post-Intervention
		BM (*n* = 56)	O (*n* = 44)	O+I (*n* = 29)		BM	O	O+I		
Positive Attributes to Online Shopping	Prices are affordable online				*p =* 1.0				*p =* 0.138	*p* = 0.02 *
	Agree/Strongly Agree	18 (69%)	16 (72%)	7 (77%)		22 (73%)	6 (43%)	6 (75%)		
	Disagree/Strongly Disagree	8 (30%)	6 (27%)	2 (22%)		8 (26%)	8 (57%)	2 (25%)		
	Quality of the food is good online				*p* = 0.63				*p* = 0.006 *	*p* = 0.57
	Agree/Strongly Agree	22 (84%)	15 (75%)	7 (88%)		24 (77%)	9 (39%)	4 (36%)		
	Disagree/Strongly Disagree	4 (16%)	5 (25%)	1 (12%)		7 (22%)	14 (60%)	7 (63%)		
	Availability of food items I like online				*p* = 0.778				*p* = 0.001 *	*p =* 0.346
	Agree/Strongly Agree	21 (72%)	13 (68%)	4 (57%)		8 (25%)	19 (73%)	5 (63%)		
	Disagree/Strongly Disagree	8 (27%)	6 (32%)	3 (42%)		24 (75%)	7 (27%)	3 (37%)		
	Access to internet				*p* = 0.60				*p* = 0.79	*p* = 0.645
	Agree/Strongly Agree	31 (100%)	29 (95%)	16 (100%)		36 (97%)	28 (97%)	16 (94%)		
	Disagree/Strongly Disagree	0	1 (5%)	0		1 (3%)	1 (3%)	1 (6%)		
	Option for delivery is available online for me				*p* = 0.087				*p* = 1.0	*p* = 0.584
	Agree/Strongly Agree	26 (57%)	18 (78%)	15 (83%)		31 (68%)	19 (70%)	10 (71%)		
	Strongly Disagree	19 (42%)	5 (21%)	3 (16%)		14 (31%)	8 (29%)	4 (28%)		
	Online shopping saves time				*p* = 0.497				*p* = 0.249	*p* = 0.197
	Agree/Strongly Agree	25 (86%)	18 (94%)	13 (81%)		36 (94%)	21 (84%)	14 (82%)		
	Disagree/Strongly Disagree	4 (13%)	1 (5%)	3 (18%)		2 (6%)	4 (16%)	3 (18%)		
Barriers to Online Shopping	Online site difficult to use				*p* = 0.103				*p* = 0.001 *	*p* = 0.001 *
	Agree/Strongly Agree	12 (28%)	8 (21%)	2 (7%)		24 (50%)	6 (17%)	2 (8%)		
	Disagree/Strongly Disagree	31 (72%)	30 (79%)	25 (93%)		24 (50%)	28 (83%)	25 (92%)		
	Search for labels takes too long				*p* = 0.336				*p* = 0.001 *	*p* = 0.008 *
	Agree/Strongly Agree	13 (35%)	8 (25%)	2 (12%)		24 (59%)	6 (25%)	2 (9%)		
	Disagree/Strongly Disagree	29 (69%)	24 (75%)	15 (88%)		17 (41%)	18 (75%)	18 (91%)		
	Online pick up times are inconvenient				*p =* 0.069				*p* = 0.005 *	*p* = 0.015 *
	Agree/Strongly Agree	20 (44%)	11 (35%)	3 (15%)		22 (56%)	6 (23%)	4 (20%)		
	Disagree/Strongly Disagree	25 (55%)	20 (65%)	17 (85%)		17 (43%)	20 (77%)	16 (80%)		
	Delivery fees make me less likely to order				*p =* 0.069				*p* = 0.475	*p* = 0.039 *
	Agree/Strongly Agree	22 (56%)	13 (46%)	11 (84%)		28 (65%)	13(50%)	11 (58%)		
	Disagree/Strongly Disagree	17 (43%)	15 (64%)	2 (15%)		15 (35%)	13 (50%)	8 (42%)		
	Minimum purchase is a barrier to ordering online				*p =* 0.293				*p* = 0.002 *	*p* = 0.7
	Agree/Strongly Agree	22 (38%)	18 (38%)	14 (18%)		28 (66%)	9 (33%)	4 (24%)		
	Disagree/Strongly Disagree	14 (61%)	11 (62%)	3 (82%)		14 (33%)	18 (66%)	13 (76%)		

^1^ Means and percentages were derived with descriptive statistics. Chi-square was used to test for differences across study arms and differences between baseline and post-intervention. * Indicates significant differences between study arms (*p* < 0.05).

## Data Availability

Data are available through the corresponding author. Data are not publicly available due to confidentiality.

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
