# Peer review of "Online Pilot Grocery Intervention among Rural and Urban Residents Aimed to Improve Purchasing Habits"

_ijerph, 2022, doi:10.3390/ijerph19020871_

Round 1
Reviewer 1 Report
This is an important topic and the study provides timely information. I do have a concern that the high number of study dropouts limits the ability to draw conclusions, as does the fact that you do not have complete information on the household food supply for all individuals (all purchases and acquisitions). I would suggest, therefore, considering this as a preliminary study that provides some highly interesting and suggestive findings, but ones that need to be confirmed with a larger, more rigorous study.
I appreciate that the authors analyzed the demographic characteristics of dropouts, but I think more study of dropouts would have been helpful. Generally, a qualitative follow up of both dropouts and study completers (perhaps a subgroup of them, to make the task more manageable) would have been helpful at this point. For dropouts, understanding reasons for dropout would inform any consideration of selection bias in the results. For completing online shoppers, interviews could have illuminated their perceived costs and benefits, suggesting how SNAP online food purchasing policies and educational communications could be crafted. Information that might have been gleaned might have included why some people quit the study and/or online shopping; why some switched between spending options; what was hard/easy about online shopping; recommendations for online shopping tips or nutrition education; whether there were other outlets they bought or acquired food from (gas stations, convenience stores (? not sure if receipts from these were submitted), fast food or other restaurants, school meals, food banks, etc).
Some specific comments:
ABSTRACT: Suggest the sentence "On average, participants in the O+I arm spent $6.84 more on F&V compared to BM arm." be rephrased to "On average reported food spending on F&V by participants.....etc" since you acknowledge you may not have complete food spending.
INTRODUCTION
Explain why online purchasing was previously unavailable through SNAP--legal requirement for face-to-face transactions. Clarify whether you mean "online shopping" to include online ordering and at-store pickup as well as online ordering and delivery. It seems like online ordering/at-store pickup could be done without a waiver if payment were concluded at pick up. That could affect availability, convenience and cost of online shopping for SNAP participants, so it's worth being clear about this.
Methods
Study Design: How many participants in BM group had no access to online shopping vs those randomized to the group? Why disparate numbers in initial O group (60) vs O+ I group (51)? Since there ended up being higher dropout in the O+I group, that was unfortunate.
F&V spending calculations: It sounds like you included mixed foods (you mention soups). Can you give some more detail on what was included--canned stews? chili or spaghetti sauce (tomato sauce a vegetable?, it is in the USDA/ARS Food Group database), vegetable lasagna or tv dinner? Also, can you give some explanation of how you assessed the share of spending that could be assigned to the F&V component of these mixed foods? Did you follow any existing method or develop one of your own? If this is lengthy, perhaps it could be an online appendix.
Data Analysis: Given the high number of dropouts, I appreciate the sensitivity analysis. However, similarity on the demographics you analyzed might be a result of the fact that your overall sample had very little variation in those areas--Table 1 shows pretty homogeneous demographics. It would be good to be able to use a method that could identify selection and correct for it (e.g. Heckman) but given your small sample size, that is likely not feasible. It is important to acknowledge the presence of unobservable that may have resulted in dropoff and therefore results driven by selection. This is where some interviews of dropouts might have shed some light.
RESULTS
Interesting that 13% of BM shoppers also shopped online. Presumably these were not the ones you had identified as having no online availability? Maybe a note on this would be helpful as other readers than myself may be surprised by the contradiction.
If "online" included both ordered online for in-store pickup and ordered online for delivery, can you provide a breakdown the percent of online purchasing falling into each category? If only "online for delivery" is included, please make that clear.
Do you have any data on the extent to which purchases were made with SNAP or the participant's cash funds? Surprising that so many of your participants were not on SNAP, according to Table 1.
Table 1. Food spending variables: Include denominator (e.g. weekly spending). Total Bill: Can you compare this figure to some published amount such as weekly spending allocated under Thrifty Food Plan, weekly food spending by low-income households (I believe the USDA Household Food Security Report may have that figure or a monthly figure)? This will help readers get a sense of the reasonableness and completeness of your data.
Findings on attitudes toward online shopping (Table 4): unfortunately, the numbers in each group are so small and so affected by dropout, that it's hard to feel confident that these are conclusive. This is where I believe a mixed-methods approach with an interview debriefing participants might have been helpful. For example, why did the O group's belief in affordability drop so much? Or did it? Of the 22 participants responding at baseline, only 14 responded post-intervention--is the change more a result of losing participants who answered yes to the question on affordability? Given that most readers will likely share my questions, some interviews that shed light on perceptions of affordability might have been helpful. Likewise, the question on whether the online site is difficult to use---it seems likely that respondents in different counties may have used different sites. So is it an issue of all online sites? some online sites? some participant characteristics? A good question for follow up, but one that can't be answered here.
Discussion
I appreciate the acknowledgement of limitations. In addition to discussion of changing food retailer landscape, you might want to discuss issue of broadband access as it pertains to needs of rural consumers. I would like to see some discussion of future research probing some of the issues you identified, such as affordability, ease of use of online sites, and tests of useful "nudge" strategies to incorporate into sites.
Reviewer 2 Report
The authors conducted an 8-week online shopping intervention to determine impact on purchase of fruits and vegetables. As the authors mentioned in the introduction, rural individuals usually cannot access online grocery shopping due to retailer limitation which was noted in this study as many rural individuals had to serve as the control group. Would have appreciated reading more about how retailers could address this issue in the discussion. Other comments to help strengthen this manuscript are:
Title: Suggest a concise method to write this title as opposed to what appears to be the objective statement. Maybe, “Online grocery intervention among rural and urban residents in improving shopping habits”?
Abstract: clarify if the participants were equally randomized into 1 of 3 interventions
Introduction: Mentioned that rural adults were unable to access online grocery shopping due to various barriers, but then in the next sentence mentioned how online shopping could improve access and dietary intake. To the reader these 2 sentences next to each other appears contradictory even though there is mention about overcoming barriers. Also, there is lack of literature within this paragraph to support that online grocery shopping leads to improved shopping behaviors/dietary habits.
Would recommend avoiding or reducing the connection between poor purchases on online shopping to increasing diet-related diseases unless that is the purpose of this study. If online shopping has the potential to create more unhealthy habits than physically grocery shopping, please include that comparison to help strengthen the argument for the need of this study.
As it is known that during the pandemic, it was difficult to purchase items physically at the grocery store. However, it would be good to include the purchases that individuals on SNAP normally make at the grocery store as there are several interventions conducted with nutrition education, product placement, etc that showed some improvements in purchases of “healthy” items when doing it physically in stores. Therefore, are the same principles being applied in this study to demonstrate that these principles not only work in the physical establishment, but also online? If so, include this information.
For the final paragraph of the introduction, suggest the 3-arms are removed and placed in the methods section.
Methods: As an inclusion criteria, did they have to be on SNAP or other indicators that they were low-income? If so, please include that information as that was mentioned in the introduction as far as those not able to really access online shopping. Additionally, beyond the capability to receive text messages, did they also have to have internet for online shopping? If so, include this information. Please also include the exclusion criteria. Expand on the randomization process- computer generated or…. For the retailers used, was it all consistently Wal-mart or Amazon to ensure the pricing of fruits/vegetables were consistent in those regions? Please clarify.
What was the targeted sample size as this appears to be more of a pilot/feasibility study? Explain based on power calculation.
Instead of indicating that the rural residents who had no access to online shopping due to the retailer and thus automatically placed in the control group, that it is worded another way to indicate that those who did not have online shopping option were considered the control group. At baseline, was there an attempt to collect their current shopping habits? If so, include that information.
There are theories that surround behavioral nudges, were any of these theories utilized? If so, please include that information. Also, describe who validated these text messages. For these messages, were there specific amounts of fruits and vegetables that they should purchase each week or was it more geared towards eating X servings of fruits/vegetables daily? Even though an example for a text message was provided, were there questions or other methods used to attempt to engage a participant? The way that particular message is written, it is difficult for the reviewer to know how the participant could engage with that information unless a simple ‘ok’ was expected.
Results: Observing the amount spent on groceries, was there also household number included and controlled for?
Discussion: There is limited information about why behavioral nudges work/do not work for the context of this study as well as why interaction was decreasing throughout the length of the intervention. Another limitation is that rural participants had to be in the control group due to limited ability to access a retail who had online shopping options.
Round 2
Reviewer 2 Report
The authors have significantly improved this manuscript, which is stronger and helps inform readers what was done in case scientists would like to replicate this study. In the first sentence of the conclusion - innovative, etc is opinions of the authors and should be reworded. Also, there is a mix between first and third person language in the methods and results, suggest being consistent.
Author Response
Reviewer Response:
The authors have significantly improved this manuscript, which is stronger and helps inform readers what was done in case scientists would like to replicate this study. In the first sentence of the conclusion - innovative, etc is opinions of the authors and should be reworded. Also, there is a mix between first and third person language in the methods and results, suggest being consistent.
Our Response:
We have reworded the first sentence of the conclusion to now read:
This pilot study provides suggestive findings related to how online shopping may improve food shopping habits but ones that need to be confirmed with a larger, more rigorous study.
We have reviewed the methods and results and made the following changes for consistency;
Changes to were; changes to have; and reviewed results related to past tense.